# Learning Generative Models with Locally Disentangled Latent Factors

## Abstract

One of the most successful techniques in generative models has been decomposing a complicated generation task into a series of simpler generation tasks. For example, generating an image at a low resolution and then learning to refine that into a high resolution image often improves results substantially. Here we explore a novel strategy for decomposing generation for complicated objects in which we first generate latent variables which describe a subset of the observed variables, and then map from these latent variables to the observed space. We show that this allows us to achieve decoupled training of complicated generative models and present both theoretical and experimental results supporting the benefit of such an approach.

## 1 Introduction

Learning useful intermediate representations in a hierarchical manner has been a driving factor in the recent success of deep learning (Goodfellow et al., 2016) (Krizhevsky et al., 2012). When ample amounts of labelled data are available, supervised learning methods are successful in learning useful intermediate representations (Zeiler & Fergus, 2014) (Nguyen et al., 2016a). However, the task is significantly more challenging in the context of unsupervised learning. One such approach to unsupervised learning is to learn a generative model of high-dimensional observed variables with low-dimensional latent variables, such that the latent variables capture the salient features of the data, which could then be used for other upstream tasks.

Recently, there has been an increased interest in learning *hierarchical* latent variable generative models. While a typical latent variable generative model uses multilayer neural networks, the generative process still assumes a single latent variable structure which generates the observed variable. The stochasticity of the observed variables has to be completely captured in a single latent vector. A *hierarchical* latent variable model posits that the data could be better explained by a sequential process with multiple latent variables, $\mathbf{z}_0 \rightarrow ... \rightarrow \mathbf{z}_{n-1} \rightarrow \mathbf{x}$. The motivation for hierarchical latent variables follows from the analogous result in supervised representation learning (Lin et al., 2017) for deterministic intermediate representations, where the authors argue that a hierarchical neural network uses capacity exponentially more efficiently.

Recent work by (Zhao et al., 2017) argues why hierarchical latent variables models are often not able to take advantage of the hierarchy, and only the lowest-level latent variables learn any useful representations. We posit that this is possibly because the vanilla hierarchical latent variable structure by itself only adds a very weak prior (that of devoting more processing to higher-level latent variables). When parameterizing the conditional distributions with powerful deep neural networks, this could admit a local optima in which all factors of variation are *sub-optimally explained* by the lowest-level latent variable. Notably, this phenomenon was also common in supervised training of deep neural networks, before (Ioffe & Szegedy, 2015) introduced *batch normalization*, which successfully disentangles the learning dynamics at each layer, as if each layer has an independent objective function.

Recent success in hierarchical latent variable models (Reed et al., 2017; Zhang et al., 2016; Nguyen et al., 2016b) contribute motivating examples of successful hierarchical latent variable models. Such approaches either use a resolution-based hierarchy or extract the hierarchy from a discriminative model trained with labelled data. These examples give sufficient evidence of the *existence* of good hierarchical latent variable models, but discover them using overly strong priors or label information.

For example, the resolution-based hierarchy is well suited to images because lower resolution images capture some factors of variation (such as objects) but discards other factors of variation (such as texture and details), giving the low and high level models distinct responsibilities. However, this decomposition is a strong prior and may not work well or apply to other types of data (for example, it is not clear how it would apply to language or video).

This motivates the need for an unsupervised method for learning hierarchical latent variables with a requisite but general prior to facilitate disentangled learning dynamics in each level.

Our proposed approach, which we call Locally Disentangled Factors (LDF), has the following desired features:

- Decoupled level-wise training objectives which significantly accelerate training.
- A graphical model based on spatial locality which aids in credit assignment, and can be thought of as a generalization to resolution-based hierarchies.
- Vastly reduced memory consumption which allows training generative models on large objects, such as videos, where this is known to be a prohibitive limitation.
- Applicable to variable-length objects, such as videos and text.

## 2 PROPOSED APPROACH

### 2.1 MOTIVATING EXAMPLE

Consider, for instance, a photograph of a busy street. Such an image has various types of stochasticity. At the highest level, consider a factor that specifies the time of day, or even the season. Such a high-level factor of variation affects the observation drastically. Every pixel could be different based on this factor. Moreover, every other factor of variation depends on this highest-level factor of variation. For example, the clothing of a person depicted in the picture, which is a mid-level latent variable, depends on the season factor. At the lowest-level, we have details such as leaves, texture, etc. Such low-level factors explain pixels in a small region.

The above example motivates us to introduce a spatial structure to the hierarchy, i.e. assign the lowest-level latent variables to only model small regions of the image. The subsequent level then models a small local group of latent variables below it, and so on. At the end, the topmost latent variable models all the latent variables below it and, thus, indirectly the whole image.

### 2.2 HIERARCHICAL LATENT VARIABLE GENERATIVE MODELS

Let $\mathbf{X} = \{\mathbf{x}^{(i)}\}_{i=1}^{N}$ be a dataset of samples of random variable $\mathbf{x}$ drawn from a distribution $\mathcal{P}(\mathbf{x})$. We now assume that the data is generated by a stochastic process defined by the Bayesian network

$$\mathbf{z}_0 \rightarrow ... \rightarrow \mathbf{z}_{n-1} \rightarrow \mathbf{x}$$

such that the joint distribution factorizes as

$$\mathcal{P}(\mathbf{z}_0, ..., \mathbf{z}_{n-1}, \mathbf{x}) = P(\mathbf{z}_0) \left( \prod_{i=1}^{n-1} \mathcal{P}(\mathbf{z}_i | \mathbf{z}_{i-1}) \right) \mathcal{P}(\mathbf{x} | \mathbf{z}_{n-1})$$

Our approach uses the adversarially learnt inference (ALI) framework (Dumoulin et al., 2017), which matches the generative process and the inference process using the generative adversarial learning framework (Goodfellow et al., 2014).

Consider the joint distribution specified by the inference process

$$\mathcal{Q}(\mathbf{z}_0, ..., \mathbf{z}_{n-1}, \mathbf{x}) = \mathcal{Q}(\mathbf{x}) \mathcal{Q}(\mathbf{z}_{n-1} | \mathbf{x}) \left( \prod_{i=1}^{n-1} \mathcal{Q}(\mathbf{z}_{i-1} | \mathbf{z}_i) \right).$$

Consider also a discriminator $\mathcal{D}_i(\mathbf{z}_i, \mathbf{z}_{i+1})$, which takes as input samples from two consecutive levels, and attempts to learn whether these samples are from the generative process or the inference process.

The task for the generative and inference processes is to converge to the same distribution such that the discriminator is not able to to differentiate between the two.

The objective is to find the saddle-point of the following minmax game for each pair of consecutive random variables $\mathbf{z}_i, \mathbf{z}_{i+1}$, where $\mathbf{z}_n$ is taken to mean $\mathbf{x}_{i+1}$.

$$\inf_{\mathcal{P}, \mathcal{Q}} \sup_{\mathcal{D}} \mathbb{E}_{\mathbf{z}_i, \mathbf{z}_{i+1} \sim \mathcal{P}} \left[ \log \mathcal{D}_i(\mathbf{z}_i, \mathbf{z}_{i+1}) \right] - \mathbb{E}_{\mathbf{z}_i, \mathbf{z}_{i+1} \sim \mathcal{Q}} \left[ \log(1 - \mathcal{D}_i(\mathbf{z}_i, \mathbf{z}_{i+1})) \right]$$

As is shown in (Goodfellow et al., 2014; Nowozin et al., 2016), the Nash equilibrium of this minimax game results in the inference and generative distributions minimizing the Jensen-Shannon Divergence which is minimized when $\mathcal{P} \approx \mathcal{Q}$, which is our desired goal.

The LDF approach can be seen as a hierarchical variant of Adversarially Learned Inference (Dumoulin et al., 2017) where each level of latent variables is constrained to follow a prior, which allows us to decouple each level.

## 3 LOCALLY DISENTANGLED FACTORS

The above is a general framework for estimating latent variable generative models. We further introduce *local connectivity* and *disentanglement* in latent variables. *Local connectivity* further introduces independence assumptions in the conditional distributions of the generative process. For simplicity let's assume that the random variable $\mathbf{z}_i$ is a $d$-dimensional vector.

**Disentanglement**    As popularized by (Kingma & Welling, 2013), *disentanglement by factorization* assumes that the conditional distribution is independently factorizable, i.e.

$$\mathcal{P}(\mathbf{z}_{i+1}|\mathbf{z}_i) = \prod_{j=1}^{d} \mathcal{P}(z_{i+1,j}|\mathbf{z}_i)$$

where $z_{i+1,j}$ is the $j^{\text{th}}$ element of the vector $\mathbf{z}_{i+1}$.

**Local Connectivity**    While local connectivity has a clear analogy to convolutions (LeCun et al., 1998) in supervised learning, we believe this is the first use of local connectivity in latent variable generative models. The local connectivity independent assumption is given by

$$\mathcal{P}(z_{i+1,j}|\mathbf{z}_i) = \mathcal{P}(z_{i+1,j}|\mathbf{z}_{i,j-p:j+p})$$

Here, $\mathbf{z}_{i,j-p:j+p}$ is a slice of the vector comprising of elements with index between $j - p$ and $j + p$.

### 3.1 IMPLEMENTATION DETAILS

For simplicity, we only considered hierarchies with two levels, although conceptually LDF could work with more than two levels. For videos, the bottom level is a single image and the top level is the collection of images across time. For images we cut the image into a grid (for example 64x64 is cut into a 4x4 grouping of 16x16 patches).

The training procedure is bottom up, starting from the leaves (observed variables), we learn disentangled latent factors at each node which also are able to reconstruct the observed variables. In theory if Adversarially Trained Inference is trained to optimality and with a stochastic decoder, the reconstructions (formed by running the inference network on real data points and then the generation network on the inferred latent states) should be exactly identical. Conceptually this works with reconstruction penalties or an ALI objective (Dumoulin et al., 2017), and in our case we use both

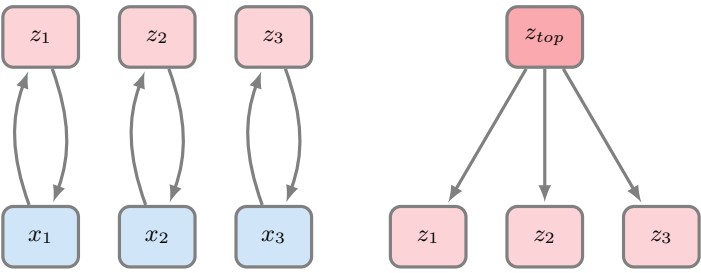

Figure 1: Diagram illustrating the training procedure for locally disentangled factors. On the left we train an ALI network on each locally disentangled factor with shared parameters. On the right we train a global generator network to produce these disentangled latent factors.

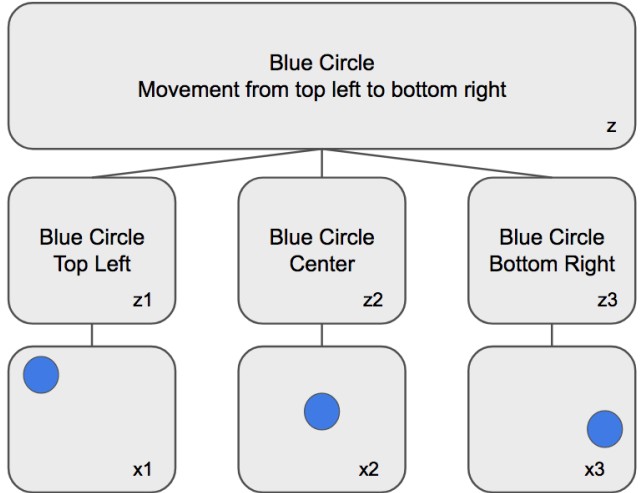

Figure 2: Illustration of a simple task where using locally disentangled factors would be expected to make training easier. In this case, we consider a video dataset where a given shape moves a little bit in each frame. Each frame can be constructed exactly at the pixel level from two independent latent factors: the shape of the object and its position. In the pixel space, a large fraction of the pixels have correlated values between different frames, but in the disentangled latent space, different independent aspects can be modeled separately (shape and movement).

the normal ALI objective as well as "shortcut reconstructions" which go through the final hidden layer before the latent variables.

In all cases we used the stabilizing regularization objective for all GAN discriminators (both higher and lower levels) (Roth et al., 2017).

## 3.2 LEARNING WITH LIMITED SAMPLES FROM THE JOINT DISTRIBUTION

Hierarchical models come with a certain statistical benefit: individual factors (e.g. a model for the frames of a video, or patches of an image) are easier to learn. For example, learning the marginal distribution for the frames of video sequences only requires data on individual frames. Importantly, those samples do not have to be drawn from the joint distribution. Then assuming that the dependencies between the factors can be described by a few parameters, one would only need a handful of samples from the joint to learn them.

We see this in practice in our experiments in Section 5; here we give a very simple argument on a toy example. Consider the class of zero-mean, $p$-variate Gaussian distributions and samples, $(X(t))_t$, corrupted by additive, isotropic Gaussian noise of variance $\sigma^2$ per covariate. Learning the distribution from samples reduces to estimating its covariance matrix, $\Sigma$. Classic results suggest that,

for a constant-rank $\Sigma$, $n = O(\sigma^4 p)$ samples from the joint, $p$-variate distribution are necessary and sufficient for recovery (Johnstone, 2001; Baik et al., 2005).

Now consider a hierarchical model that splits its $p$ variables into $k$ blocks, $X_i(t)$, for $i \in 1 \ldots k$. The covariance matrix can be broken up into its block components, $\Sigma_{ij} \in \mathbb{R}^m$, where $m = \left(\frac{p}{k}\right)$. All the parameters on the main block diagonal, $\Sigma_{ii}$ can be learned using $k$ sets of $O(\sigma^4 p/k)$ samples from the $(p/k)$-variate marginals. Now if cross-covariances are modeled approximately using a parametric model or trainable upper level, like in our experiments, we will only need enough samples from the joint to learn those parameters.

## 4 RELATED WORK

**Resolution-based Hierarchy**

The resolution hierarchy approach involves initially generating at a low resolution and then generating at higher resolutions while conditioning on the lower resolution and it has had great success in the generative models literature (Zhang et al., 2016; Reed et al., 2017; Denton et al., 2015). This is distinct from LDF but shares a closely related motivation. The resolution hierarchies approach decomposes generation into multiple stages, but the content of the higher levels is fixed (it is the downsampled, lower resolution version of the original image). This can simplify the task of training because different stages in generation become responsible for different factors of variation: generating low resolution images requires knowledge about the general shape and location of objects, whereas generating high resolution images conditioned on the low resolution images requires greater knowledge about the fine-grained details of images. In LDF, the higher levels of the hierarchy are learned to be representations which capture the details at the lower levels but are disentangled or nearly disentangled. Because the latent factors in LDF are supposed to capture all of the details at the lower level, it is possible to do decoupled training across different local factors, whereas this is not generally possible with Resolution-based Hierarchies because downsampling can remove details (for example, drawing a person's eye color needs to be done the same way in the two eyes in a face image, and this detail of eye color may not be present in a lower resolution version of the image). In addition resolution-based downsampling may perform poorly or not be applicable to certain types of data - for example it is not clear how it would be defined for language data.

**Synthetic Gradients**

Jaderberg et al. (2016) introduced the idea of improving the scalability of training deep networks by decoupling the computations for making an update to the parameters of each layer. The synthetic gradients approach trains a network for each layer which takes states of that layer as input and estimates the gradient of the loss with respect to that layer. While this still requires computing gradients through the entire network, additional updates can be made in a decoupled fashion by using these synthetic gradient modules. Like Synthetic Gradients, Locally Disentangled Factors allows for decoupled updating. However, unlike Synthetic Gradients, Locally Disentangled Factors relies on the statistical properties of the data to achieve decoupled training - more specifically it assumes that the selected local regions can be efficiently described by disentangled latent factors. On the other hand, the decoupling in synthetic gradients is achieved when the synthetic gradient modules are able to successfully predict the full gradients (or at least produce gradients that work equally well for training). At a first glance, these two approaches are complementary, but exactly how they relate could be an interesting topic for future work.

**Adversarial Message Passing**

Karaletsos (2016) proposed to perform training and inference in graphical models by using local discriminators which only see the variables in local factors. This is related to LDF in that both use discriminators which only see a subset of the variables, yet differs in that no part of the Adversarial Message Passing objective explicitly encourages the lower level latent variables to learn disentangled representations.

**Learning Hierarchical Features from Generative Models**

Zhao et al. (2017) presents a critique of hierarchical latent variable models, which essentially argues that the value of having multiple layers in the hierarchy is limited, because sampling from the joint

distribution can be achieved by doing blocked Gibbs sampling between the observed variables and the lowest level of latent variables. They demonstrate that this prevents hierarchical variational autoencoders from learning useful latent variables from the higher levels. An important assumption in their critique is that the Markov chain that is sampled on the lower levels is ergodic, which is generally the case when the noise injected in the lower levels of the hierarchy is Gaussian. However, since the lower level of our hierarchy is an ALI network, we can make this sampling process potentially non-ergodic by making $q(z_i|x_i)$ and $p(x_i|z_i)$ deterministic.

As a further illustration of this limitation in the critique by Zhao et al. (2017), consider a hierarchical model which first generates faces at a 256x256 resolution and then generates, conditioned on that, at a resolution of 512x512. If one were to do blocked Gibbs sampling between the 256x256 resolution and 512x512 resolution on faces for example, the sampling process would be non-ergodic and the chain would not mix (i.e. running a sampling chain between 256x256 and 512x512 resolution images of a face will never significantly change the identity of the face), which means that the model could be getting value out of having a deep hierarchical model. Indeed it has been observed that similar models have been highly successful (Zhang et al., 2016), despite hypothetically being subject to the Zhao et al. (2017) critique if the blocked Gibbs sampling on the lower two levels were ergodic.

## 5 Experiments

The main goal of our experiments is to demonstrate that LDF is able to successfully learn the dependencies in the data, even though the training is localized: no discriminator covers all of the variables and no gradient flows through the entire graph (the higher level and lower level models are updated locally). We demonstrate that this can be done successfully on both image generation, where we divide the image into quadrants with distinct latent factors, and on video generation, where we divide a video with five frames into single images each with distinct latent factors.

### 5.1 Image Generation

We evaluate our approach on image generation on the CIFAR dataset. We selected this dataset primarily because the Inception Score method (Salimans et al., 2016) provides a way of quantitatively evaluating results. Each image is 32x32, and for LDF, we divide this into a 2x2 grid giving us four lower level latent segments. The lower level models share parameters across all positions.

### 5.2 Video Generation

We evaluated our model on unconditional video generation by considering the Pacman video dataset collected by (Cooper, 2017). This dataset consists of 20000 videos each containing 5 adjacent frames. The 32x32 video patches are shown in figure 5 and were selected randomly from full Pacman game screens but filtered so that most of the clips contain movement. The Pacman dataset was selected because it is visually simple and has predictable motion (the movement of Pacman and the ghosts). Additionally, because the background never changes, the ability of the model to keep a constant background is a test of the ability of LDF to learn the relationship between different steps.

For LDF, each frame's lower level model (generator, inference network, and discriminator) are convolutional neural networks with an architecture similar to that used in (Dumoulin et al., 2017). The higher level models (generator and discriminator) are fully-connected MLPs. For the baseline model referenced in the Pacman figures, we train a simple GAN which has the frames stacked together across the filters with each video treated as if it were an image with 15 channels (5 frames and 3 colors).

### 5.3 Evidence that Local Disentangling Simplifies Training

We conducted experiments to demonstrate that using LDF simplifies the training procedure. We did this in two ways. First, we showed that using LDF trains faster than a joint training baseline (a GAN where the discriminator sees all of the visible variables directly and the generator outputs all variables in the visible space directly). This is shown in 7. Second, we showed that in the case where all of the lower level samples are available (i.e. all of the frames in the video dataset) but only

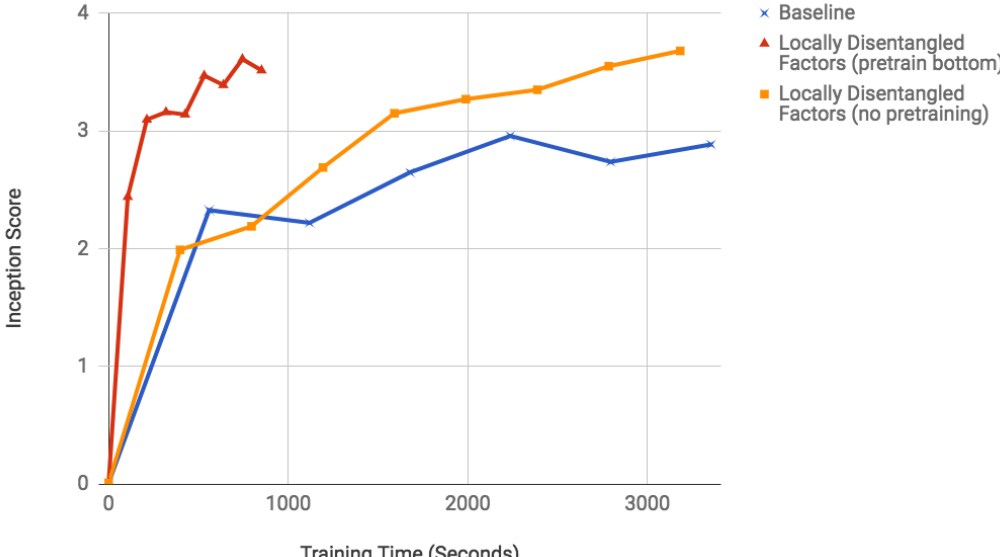

Figure 3: Inception Scores on CIFAR after 8 epochs of training with three different models. The baseline (blue) is a GAN where the discriminator sees all of the pixels and the generator produces the entire image. LDF with a pre-trained lower level (red) and LDF with training starting from scratch all train faster than the baseline GAN.

a few full samples are available (actual video sequences), training with LDF is more successful. This is shown in Fig. 6 where both LDF and the baseline joint model are only trained on 256 video sequences.

## 6  CONCLUSION

We have proposed Locally Disentangled Factors (LDF), a powerful new approach to decomposing the training of generative models. We have shown that LDF is able to successfully generate joint distributions over complicated objects, even though the discriminators and gradient flow are entirely local within the hierarchy. We have also shown that this allows for decoupled training and improved ability to learn from small amounts of data from the joint distribution. While our method assumes a more general prior than resolution-hierarchy style approaches, it still leaves the decision of what the local factors would be on the practitioner. Finding methods that enjoy the same computational and sample-complexity benefits with fewer assumptions about the data is an interesting research direction.

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

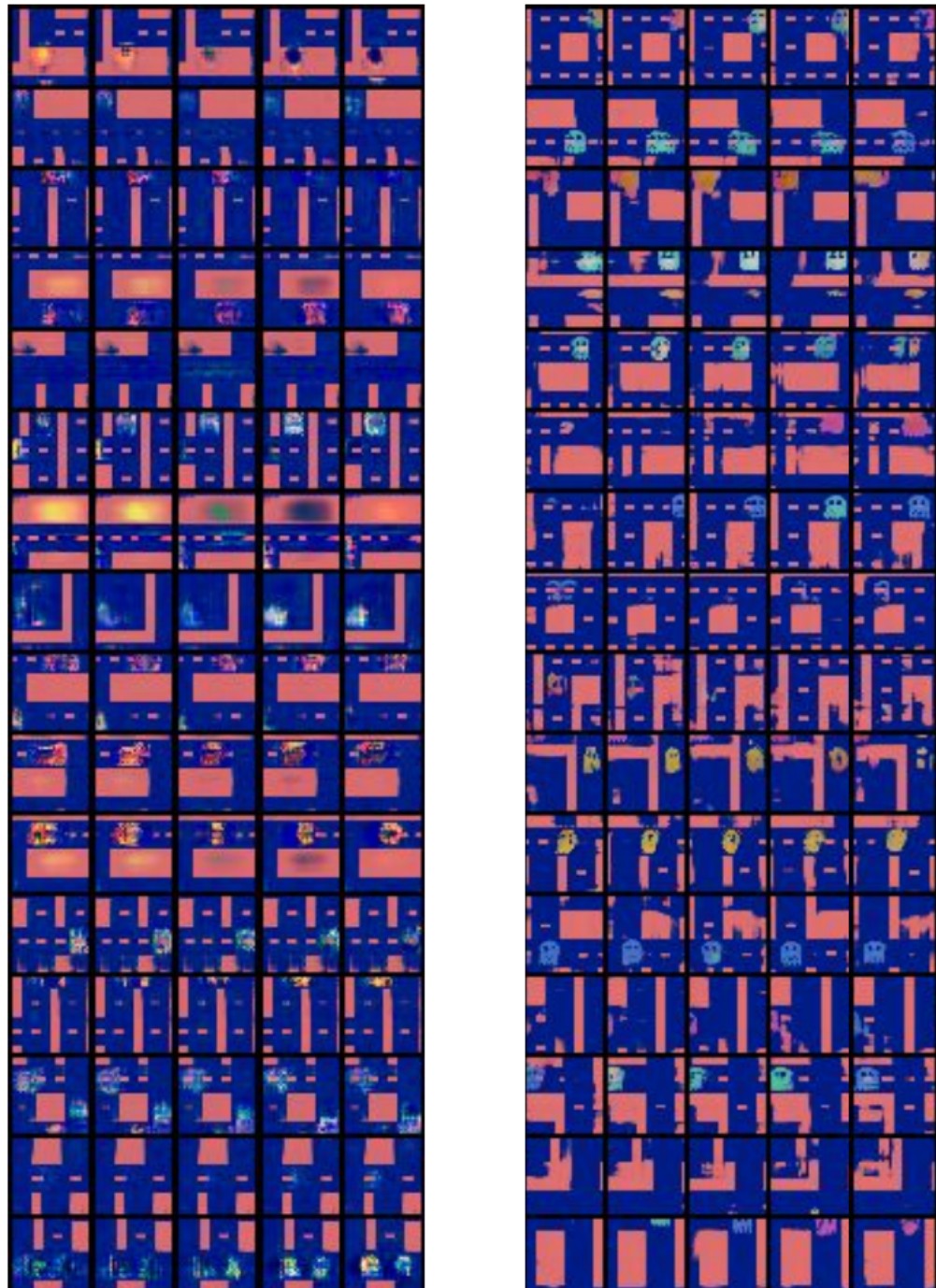

Figure 4: Unconditioned video generation samples trained using a joint model where a discriminator considers all frames (left) and locally disentangled factors (right).

Ian Goodfellow, Jean Pouget-Abadie, Mehdi Mirza, Bing Xu, David Warde-Farley, Sherjil Ozair, Aaron Courville, and Yoshua Bengio. Generative adversarial nets. In *Advances in neural information processing systems*, pp. 2672–2680, 2014.

Ian Goodfellow, Yoshua Bengio, and Aaron Courville. *Deep Learning*. MIT Press, 2016. http://www.deeplearningbook.org.

Sergey Ioffe and Christian Szegedy. Batch normalization: Accelerating deep network training by reducing internal covariate shift. In *International Conference on Machine Learning*, pp. 448–456, 2015.

Max Jaderberg, Wojciech Marian Czarnecki, Simon Osindero, Oriol Vinyals, Alex Graves, and Koray Kavukcuoglu. Decoupled neural interfaces using synthetic gradients. *CoRR*, abs/1608.05343, 2016. URL http://arxiv.org/abs/1608.05343.

Iain M Johnstone. On the distribution of the largest eigenvalue in principal components analysis. *Annals of statistics*, pp. 295–327, 2001.

T. Karaletsos. Adversarial Message Passing For Graphical Models. *ArXiv e-prints*, December 2016.

Diederik P Kingma and Max Welling. Auto-encoding variational bayes. *arXiv preprint arXiv:1312.6114*, 2013.

Alex Krizhevsky, Ilya Sutskever, and Geoffrey E Hinton. Imagenet classification with deep convolutional neural networks. In *Advances in neural information processing systems*, pp. 1097–1105, 2012.

Yann LeCun, Léon Bottou, Yoshua Bengio, and Patrick Haffner. Gradient-based learning applied to document recognition. *Proceedings of the IEEE*, 86(11):2278–2324, 1998.

Henry W Lin, Max Tegmark, and David Rolnick. Why does deep and cheap learning work so well? *Journal of Statistical Physics*, 168(6):1223–1247, 2017.

Anh Nguyen, Alexey Dosovitskiy, Jason Yosinski, Thomas Brox, and Jeff Clune. Synthesizing the preferred inputs for neurons in neural networks via deep generator networks. In *Advances in Neural Information Processing Systems*, pp. 3387–3395, 2016a.

Anh Nguyen, Jason Yosinski, Yoshua Bengio, Alexey Dosovitskiy, and Jeff Clune. Plug & play generative networks: Conditional iterative generation of images in latent space. *arXiv preprint arXiv:1612.00005*, 2016b.

Sebastian Nowozin, Botond Cseke, and Ryota Tomioka. f-gan: Training generative neural samplers using variational divergence minimization. In *Advances in Neural Information Processing Systems*, pp. 271–279, 2016.

Scott Reed, Aäron van den Oord, Nal Kalchbrenner, Sergio Gómez Colmenarejo, Ziyu Wang, Dan Belov, and Nando de Freitas. Parallel multiscale autoregressive density estimation. *arXiv preprint arXiv:1703.03664*, 2017.

Kevin Roth, Aurélien Lucchi, Sebastian Nowozin, and Thomas Hofmann. Stabilizing training of generative adversarial networks through regularization. *CoRR*, abs/1705.09367, 2017. URL http://arxiv.org/abs/1705.09367.

Tim Salimans, Ian J. Goodfellow, Wojciech Zaremba, Vicki Cheung, Alec Radford, and Xi Chen. Improved techniques for training gans. *CoRR*, abs/1606.03498, 2016. URL http://arxiv.org/abs/1606.03498.

Matthew D Zeiler and Rob Fergus. Visualizing and understanding convolutional networks. In *European conference on computer vision*, pp. 818–833. Springer, 2014.

Han Zhang, Tao Xu, Hongsheng Li, Shaoting Zhang, Xiaolei Huang, Xiaogang Wang, and Dimitris N. Metaxas. Stackgan: Text to photo-realistic image synthesis with stacked generative adversarial networks. *CoRR*, abs/1612.03242, 2016. URL http://arxiv.org/abs/1612.03242.

Shengjia Zhao, Jiaming Song, and Stefano Ermon. Learning hierarchical features from generative models. *CoRR*, abs/1702.08396, 2017. URL http://arxiv.org/abs/1702.08396.

APPENDIX

REAL VIDEO SEQUENCES

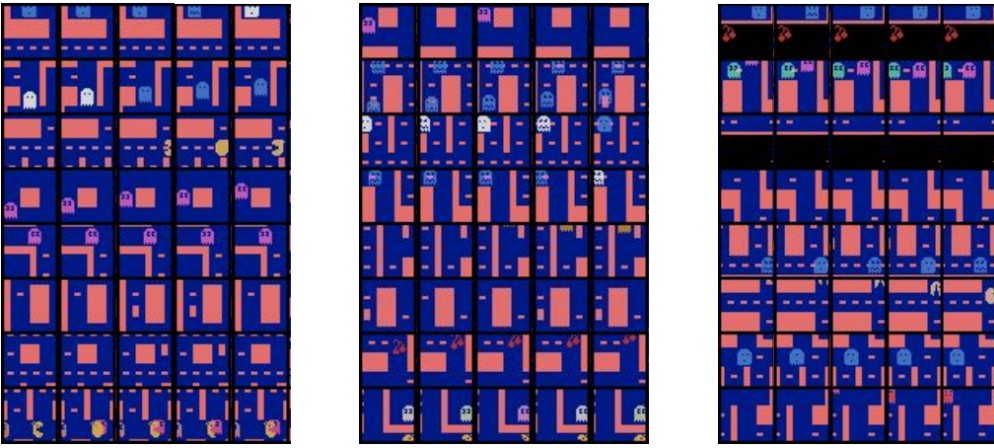

Figure 5: Real video sequences from the Pacman dataset (Cooper, 2017).

MODEL SAMPLES

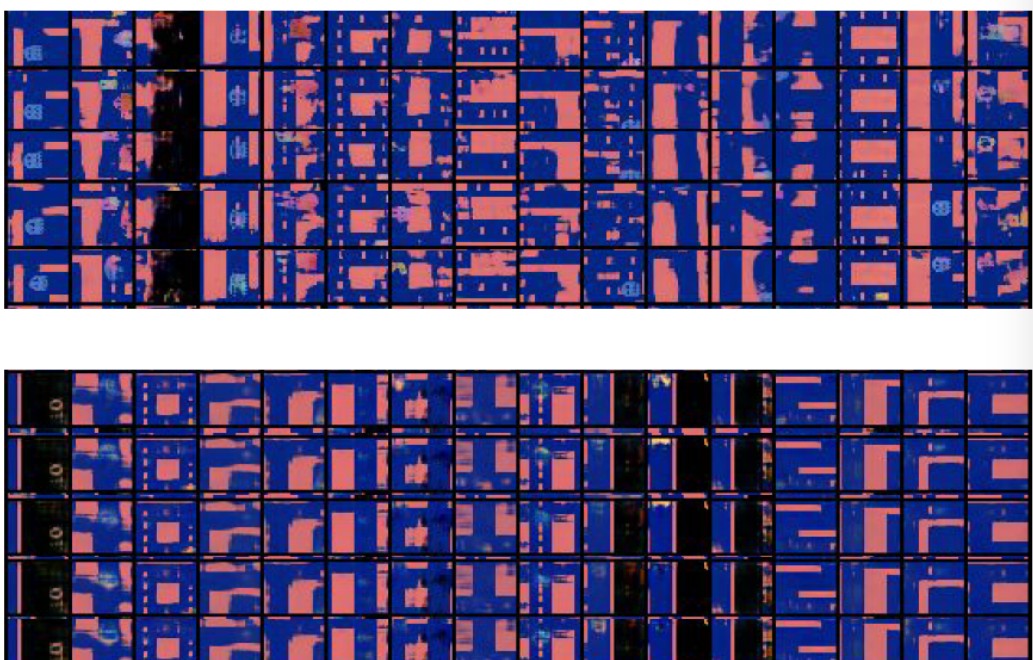

Figure 6: Unconditioned video generation samples where only 256 video samples are available (but the locally disentangled factors model can use all of the individual frames to train the local model). Baseline with joint generation model (left) and ours (right). Both models were trained for the same number of updates (five thousands epochs).

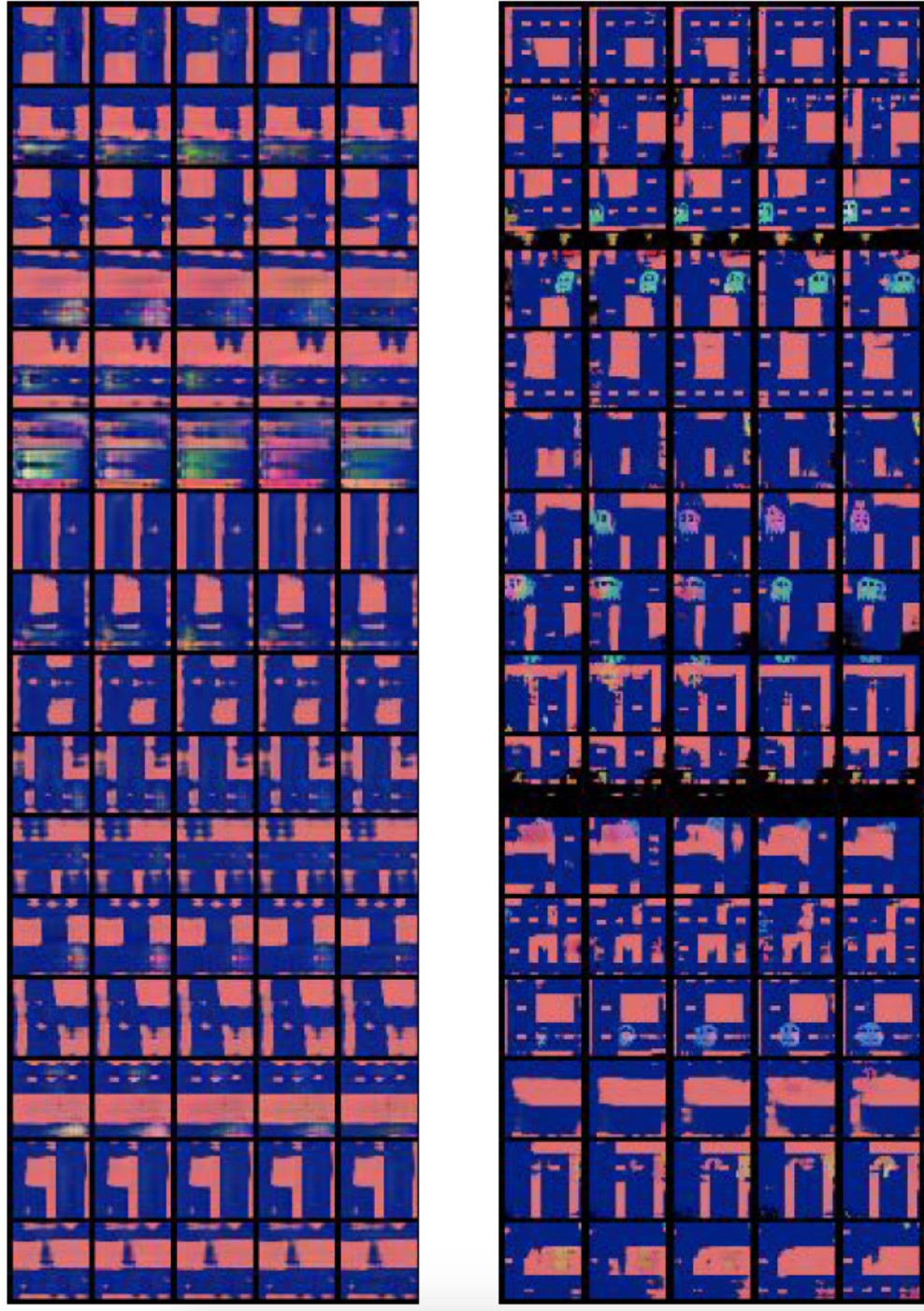

Figure 7: Unconditioned video generation samples with only 21 minutes of wall-clock training time using full video sequences. Baseline (left) and locally disentangled factors (right). The baseline runs for 10 epochs.

