# OpenReview forum: "Learning Generative Models with Locally Disentangled Latent Factors"
_ICLR.cc/2018/Conference — Reject_

### Official Review · AnonReviewer3 · 2017-11-27
**This work is incomplete and does not worth publishing with its current quality.**

**Rating:** 4
**Confidence:** 4

**Review:**

This paper proposed a method called Locally Disentangled Factors for hierarchical latent variable generative model, which can be seen as a hierarchical variant of Adversarially Learned Inference (Dumoulin el atl. 2017). The idea seems to be a valid variant, however, the quality of the paper is not good. The introduction and related works sections read well, but the rest of the paper has not been written well. More specifically, the content in section 3 and experiment section is messy. Also the experiments have not been conducted thoroughly, and the results and the interpretation of the results are not complete.

Introduction:
Although in introduction the author discussed a lot of works on hierarchical latent variable model and some motivating examples, after reading it the reviewer has absolutely no idea what the paper is about (except hierarchical latent variable model), what is the motivation, what is the general idea, what is the contribution of the paper. Only after carefully reading the detailed implementation in section 3.1 and section 5, did I realize that what the authors are actually doing is to use N variables to model N different parts of the observation, and one higher level variable to model the N variables. The paper should really more precisly state what the idea is throughout the paper, instead of causing confusion and ambiguity.

Section 3:
1. The concepts of "disentanglment" and "local connectivity" are really unnecessary and confusing. First, the whole paper and experiments has nothing to do with "local connectivity". Even though you might have the intention to propose the idea, you didn't show any support for the idea. Second, what you actually did is to use top level variable to generate N latent variables. That could hardly called "disentanglement". The mean field factorization in (Kingma & Welling 2013) is on the inference side (Q not P), and as found out in literature, it could not achieve disentanglement.

2. In section 3.2, I understand that you want to say the hierarchical model may require less data sample. But, here you are really off-topic. It would be much better if you can relate to the proposed method, and state how it may require less data.

3. Section 3.1 is more important, and is really major part of your method. Therefore, it need more extensive discussion and emphasis.

Experiment:
This section is really bad.
1. Since in the introduction and related works, there are already so many hierarchical latent variable model listed, the baseline methods should really not just vanilla GAN, but hierarchical latent variable models, such as the Hierachical VAE, Variational Ladder Autoencoder in (Zhao et al. 2017), ALI (not hierarchical, but should be a baseline) in (Dumoulin et al. 2017), etc.

2. Since currently there is still no standard way to evaluate the quality of image generation, by giving only inception score, we can really not judge whether it is good or not. You need to give more metrics, or generation examples, recontruction examples, and so on. And equally importantly, compare and discuss about the results. Not just leave it there.

3. For section 5.2, similar problems as above exist. Baseline methods might be insufficient. The paper only shows several examples, and the reviewer cannot draw any conclusion about it. Nor does the paper discuss any of the results.

4. Section 5.3, second to the last line, typo: "This is shown in 7". Also this result is not available.

5. More importantly, some experiments should be conducted to explicitly show the validity of the proposed hierarchical latent model idea. Show that it exists and works by some experiment explicitly.

Another suggestion the review would like to make is that, instead of proposing the general framework in section 2, it would be better to propose the hierarchical model in the context of section 3.1. That is, instead of saying z_0 -> z_1 ->... ->x, what the paper and experiment is really about is z_0 -> z_{1,1}, z_{1,2} ... z_{1,N} -> x_{1}, x_{2},...,x_{N}, where z_{1,1...N} are distinct variables. How section 2 is related to the learning of this might be concatenating these N distinct variables into one (if that's what you mean). Talking about the joint distribution and inference process in this way might more align with your idea. Also, the paper actually only deals with 2 level. It seems to me that it's meaningless to generalize to n levels in section 2, since you do not have any support of it.

In conclusion, the reviewer thinks that this work is incomplete and does not worth publishing with its current quality.
==============================================================
The reviewer read the response from the authors. However, I do not think the authors resolved the issues I mentioned. And I am still not convinced by the quality of the paper. I would say the idea is not bad, but the paper is still not well-prepared. So I do not change my decision.

---

> ### Author Response · Authors · 2018-01-05
> **Response**
>
> Thank you for your interesting response.
>
> "1. The concepts of "disentanglment" and "local connectivity" are really unnecessary and confusing. First, the whole paper and experiments has nothing to do with "local connectivity". Even though you might have the intention to propose the idea, you didn't show any support for the idea. Second, what you actually did is to use top level variable to generate N latent variables. That could hardly called "disentanglement". The mean field factorization in (Kingma & Welling 2013) is on the inference side (Q not P), and as found out in literature, it could not achieve disentanglement."
>
> So ultimately our goal was to learn a local representation for a part of the example which simplifies its structure as much as possible while having a 1:1 mapping with raw data for that part of the example.  One can imagine specific types of data for which this should be possible.  I think that if the disentanglement isn't perfect, it just lowers the potential benefit of our model, but it could still help.
>
> "2. Since currently there is still no standard way to evaluate the quality of image generation, by giving only inception score, we can really not judge whether it is good or not. You need to give more metrics, or generation examples, recontruction examples, and so on. And equally importantly, compare and discuss about the results. Not just leave it there."
>
> I think that Inception scores, perhaps along with FID, are reasonable to use.  However we agree that we definitely need to have a stronger baseline.
>
> However I do think that showing faster convergence here is a compelling result.

---

### Official Review · AnonReviewer1 · 2017-11-28
**Hierarchical (2-level) latent variable model based on ALI from [Dumoulin et al, ICLR'16]**

**Rating:** 6
**Confidence:** 3

**Review:**

The paper investigates the potential of hierarchical latent variable models for generating images and image sequences. The paper relies on the ALI model from [Dumoulin et al, ICLR'16] as the main building block. The main innovation in the paper is to propose to train several ALI models stacked on top of each other to create a hierarchical representation of the data. The proposed hierarchical model is trained in stages. First stage is an original ALI model as in [Dumoulin et al]. Each subsequent stage is constructed by using the Z variables from the previous stage as the target data to be generated.

The paper constructs models for generatation of images and image sequences. The model for images is a 2-level ALI. The first level is similar to PatchGAN from [1] but is trained as an ALI model. The second layer is another ALI that is trained to generate latent variables from the first layer.

[1] Isola et al. Image-to-Image Translation with Conditional Adversarial Networks, CVPR'17

In the the model for image sequences the hierarchy is somewhat different. The top layer is directly generating images and not patches as in the image-generating model.

Summary: I think this paper presents a direct and somewhat straightforward extension of ALI. Therefore the novelty is limited. I think the paper would be stronger if it (1) demonstrated improvements when compared to ALI and (2) showed advantages of hierarchical training on other datasets, not just the somewhat simple datasets like CIFAR and Pacman.

Other comments / questions:

- baseline should probably be 1-level ALI from [Dumoulin et al.]. I believe in the moment the baseline is a standard GAN.

- I think the paper would be stronger if it directly reproduced the experiments from [Dumoulin et al.] and showed how hierarchy compares to standard ALI without hierarchy.

- the reference Isola et al. [1] should ideally be cited since the model for image genration is similar to PatchGAN in [1]

- Why is the video model in this paper not directly extending the image model? Is it due to limitation of the implementation or direclty extending the iamge model didn't work?

---

> ### Author Response · Authors · 2018-01-05
> **Response**
>
> "- baseline should probably be 1-level ALI from [Dumoulin et al.]. I believe in the moment the baseline is a standard GAN."
>
> This is a fair point, although ALI did not dramatically outperform the standard GAN in terms of generation quality, for example, in terms of inception score.
>
> "- the reference Isola et al. [1] should ideally be cited since the model for image genration is similar to PatchGAN in [1]"
>
> That's a fair point.  PatchGAN is different from our approach, but would serve as a reasonable baseline.
>
> "Summary: I think this paper presents a direct and somewhat straightforward extension of ALI. Therefore the novelty is limited."
>
> I don't agree with this.  Learning generative models which learn joints over larger and more complex objects is an important direction.  For example, learning a joint distribution over a complete day of video or audio data.  With standard approaches, this quickly becomes computationally intractable.  Only a few approaches have been proposed to deal with this issue.  To our knowledge, synthetic gradients and UORO are the most prominent.  The Locally Disentangled Factors approach, while still in its infancy, could be an important method in this area.

---

### Official Review · AnonReviewer2 · 2017-12-02
**A hierarchical extension of ALI. Not well-prepared paper**

**Rating:** 3
**Confidence:** 4

**Review:**

Training GAN in a hierarchical optimization schedule shows promising performance recently (e.g. Zhao et al., 2016). However, these works utilize the prior knowledge of the data (e.g. image) and it's hard to generalize it to other data types (e.g. text). The paper aims to learn these hierarchies directly instead of designing by human. However, several parts are missing and not well-explained. Also, many claims in paper are not proved properly by theory results or empirical results.

(1) It is not clear to me how to train the proposed algorithm. My understanding is train a simple ALI, then using the learned latent as the input and train the new layer. Do the authors use a separate training ? or a joint training algorithms. The authors should provide a more clear and rigorous objective function. It would be even better to have a pseudo code.

(2) In abstract, the authors claim the theoretical results are provided. I am not sure whether it is sec 3.2 The claims is not clear and limited. For example, what's the theory statement of [Johnsone 200; Baik 2005]. What is the error measure used in the paper? For different error, the matrix concentration bound might be different. Also, the union bound discussed in sec 3.2 is also problematic. Lats, for using simple standard GAN to learn mixture of Gaussian, the rigorous theory result doesn't seem easy (e.g. [1])  The author should strive for this results if they want to claim any theory guarantee.

(3) The experiments part is not complete. The experiment settings are not described clearly. Therefore, it is hard to justify whether the proposed algorithm is really useful based on Fig 3. Also, the authors claims it is applicable to text data in Section 1, this part is missing in the experiment. Also, the idea of "local" disentangled LV is not well justified to be useful.

[1] On the limitations of first order approximation in GAN dynamics, ICLR 2018 under review

---

> ### Author Response · Authors · 2018-01-05
> **Response**
>
> "(1) It is not clear to me how to train the proposed algorithm. My understanding is train a simple ALI, then using the learned latent as the input and train the new layer. Do the authors use a separate training ? or a joint training algorithms. The authors should provide a more clear and rigorous objective function. It would be even better to have a pseudo code. "
>
> Our method uses a separate decoupled training objective which trains the higher level module after the lower level has finished training.  We agree that having pseudocode could make this clearer.
>
> "(3) The experiments part is not complete. The experiment settings are not described clearly. Therefore, it is hard to justify whether the proposed algorithm is really useful based on Fig 3. "
>
> The main goal of our experiments is to show that exploiting the decoupling from locally disentangled factors can allow for faster training and higher capacity models.  Our inception scores on MNIST provide some evidence for the former and our video generation results provide some evidence for the latter.
>
> "Also, the idea of "local" disentangled LV is not well justified to be useful."
>
> If the data generating process actually uses locally disentangled factors, then I think the benefit is fairly apparent, in that the complexity of the learning task is greatly simplified.  Whether this actually occurs in practice is an interesting open question.

---

### Decision · Program_Chairs · 2018-01-29
**ICLR 2018 Conference Acceptance Decision**

**Decision:**

Reject

**Comment:**

Reviewers recognize the proposed method of hierarchical extension to ALI to be potentially novel and interesting but have expressed strong concerns on the experiments section. The paper also needs to have comparisons with relevant hierarchical generative model baselines. Not suitable for publication in its current form.